# Normal locomotion in zebrafish lacking the sodium channel NaV1.4 suggests that the need for muscle action potentials is not universal

**Chifumi Akiyama, Souhei Sakata** ⓘ *, **Fumihito Ono**

Department of Physiology, Division of Life Sciences, Faculty of Medicine, Osaka Medical and Pharmaceutical University, Osaka, Japan

* souhei.sakata@ompu.ac.jp

## Abstract

Extensive studies over decades have firmly established the concept that action potentials (APs) in muscles are indispensable for muscle contraction. To re-examine the significance of APs, we generated zebrafish lacking APs by editing the *scn4aa* and *scn4ab* genes, which together encode NaV1.4 (NaVDKO), using the CRISPR-Cas9 system. Surprisingly, the escape response of NaVDKOs to tactile stimuli, both in the embryonic and adult stages, was indistinguishable from that of wild-type (WT) fish. $Ca^{2+}$ imaging using the calcium indicator protein GCaMP revealed that myofibers isolated from WT fish could be excited by the application of acetylcholine (ACh), even in the presence of tetrodotoxin (TTX) indicating that NaVs are dispensable for skeletal muscle contraction in zebrafish. Mathematical simulations showed that the end-plate potential was able to elicit a change in membrane potential large enough to activate the dihydropyridine receptors of the entire muscle fiber owing to the small fiber size and the disseminated distribution of neuromuscular synapses in both adults and embryos. Our data demonstrate that NaVs are not essential for muscle contraction in zebrafish and that the physiological significance of NaV1.4 in muscle is not uniform across vertebrates.

## Introduction

Voltage-gated sodium channels (NaVs) are expressed in the brain, skeletal muscles, and heart and are considered crucial for membrane excitability. Electric excitability, that is, the action potential (AP), is coupled with the functionality of these tissues. NaVs are instrumental in generating APs [1].

Sodium channels are generally composed of alpha and beta subunits. The alpha subunit protein has a voltage sensor and pore domain, which comprise a structural passageway for sodium ions across the plasma membrane. Beta subunits are non-covalently associated with alpha subunits and modulate the function of alpha subunits. Beta subunits have also been implicated in cell adhesion, migration, and aggregation [2,3].

**Data availability statement:** Models used in NEURON software are available in https://doi.org/10.17605/OSF.IO/T8HSF

**Funding:** This work was supported by the intramural research program of the Osaka Medical and Pharmaceutical University to F.O. and by KAKENHI (JP22K06837 to S.S.) from the Japan Society for the Promotion of Science (JSPS). The funders had no role in study design, data collection and analysis, decision to publish, or preparation of the manuscript.

**Competing interests:** The authors have declared that no competing interests exist.

The mammalian NaV gene family for alpha subunits is comprised of NaV1.1–1.9 (S1 Table). The expression of these genes is distinct among tissues. NaV1.1–1.3 are mainly expressed in the neurons of the central nervous system. NaV1.4, NaV1.5, and NaV1.6–1.9 are distributed in skeletal muscles, cardiac myocytes, and dorsal root ganglion neurons, respectively. The non-ionic conductive modulatory beta subunits are encoded by β1–4 genes.

Several historical studies have uncovered the mechanisms of the excitation-contraction coupling in skeletal muscles. The excitation of motor neurons evokes an end-plate potential (EPP). NaV1.4 amplifies the post-synaptic EPPs to generate APs. AP propagates without attenuation and stimulates the dihydropyridine receptor (DHPR) in transverse tubules (T-tubules). Activation of DHPR results in the release of $Ca^{2+}$ from the sarcoplasmic reticulum (SR), allowing muscle contraction. Mutations in *SCN4A* gene encoding NaV1.4 (S1 Table) causes skeletal muscle disorders in humans. The gain-of-function mutations lead to paramyotonia congenita or hyperkalemic periodic paralysis depending on the locations of the missense mutation, whereas several loss-of-function mutations lead to congenital myasthenia [4–7]. In mice, NaV1.4 null is neonatally lethal [8]. These findings are compatible with the notion that normal AP generated by NaV1.4 is essential for muscle functionality.

Zebrafish possess several advantages for studying human diseases, including the availability of genetic resources and ease of genetic manipulation, as well as multiple applicable techniques, such as live imaging of intact animals and large-scale chemical screening. Zebrafish models have revealed the molecular and cellular mechanisms of devastating diseases, such as muscular dystrophies and amyotrophic lateral sclerosis (ALS) [9–11].

However, some features are not conserved between mammals and zebrafish. Slow-twitch fibers, one of the two major types of muscle fibers found in the vertebrate skeletal system, are distributed as a mixture of fast-twitch fibers in the muscle tissues of mammalian skeletal muscles, which is not the case in zebrafish.

In teleosts, including zebrafish, slow fibers form the superficial layer, whereas fast fibers are located in the inner layers of the trunk. Moreover, the neuromuscular junctions (NMJ) in mammalian muscles are located in a specific region called the innervation zone, whereas those in zebrafish slow muscle are located at both ends of the fibers and those in fast fibers are disseminated along the length of the fibers [12].

In this study, we generated *scn4aa* and *scn4ab* double knockout zebrafish (NaVDKO). The *scn4aa* and *scn4ab* genes together encode NaV1.4. Despite the absence of NaV currents in the trunk muscles, the swimming capability of NaVDKO embryos was indistinguishable from that of wild-type (WT) fish. Gene knockout was not observed in the adults. Mathematical simulations revealed the possibility that the EPP alone can activate DHPRs and cause full contraction of muscle fibers in both the embryos and adults. Our collective results show that NaV1.4s is dispensable for the normal functionality of the trunk muscle in zebrafish and that the physiological significance of NaV1.4 in muscles is not universal across vertebrates.

## Methods

### Ethics statement

All experiments using zebrafish were reviewed and approved by the university's Institutional Animal Care and Use Committee (#AM23-026).

**Zebrafish maintenance.** Zebrafish (*Danio rerio*) were maintained in the self-circulating systems at 28 °C and the normal day–night cycle (14 h light/ 10 h dark) at Osaka Medical and Pharmaceutical University.

### Genome editing

*Scn4aa* and *scn4ab* knockout lines were generated using CRISPR-Cas9. For the synthesis of guide RNA (gRNA), oligonucleotide pairs, 5′- TAGGTGGGACTGGCCAACGTTC-3′ and 5′-AAACGAACGTTGGCCAGTCCCA-3′ for *scn4aa*, and 5′-TAGGTGGCCGAAATCCTAATTA-3′ and 5′-AAACTAATTAGGATTTCGGCCA-3′ for *scn4ab* were annealed and ligated with DR274 (Addgene), respectively. The template DNAs for the gRNA synthesis were generated using PCR with the forward (5′-GGTCAGTATTGAGCCTCAGG-3′) and backward (5′-AAAGCACCGACTCGGTGCCA-3′) primers. The gRNA sequences were transcribed using the MEGAscript T7 Transcription Kit (Thermo Fisher Scientific). To synthesize the Cas9 mRNA, pCS2-Cas9 (Addgene) was linearized using NotI (New England Biolabs) and transcribed using the mMESSAGE mMACHINE T7 ULTRA Kit (Thermo Fisher Scientific). Cas9 mRNA (100 ng/μL) and gRNA (100 ng/μL) for *scn4aa* and *scn4ab* were co-injected into single-cell stage embryos. The CRISPR targeted sequence was GTTGGGACTGGCCAACGTTCAGG for *scn4aa* and GCTGGCCGAAATCCTAATTATGG for *scn4ab*. F1 fish were identified by sequencing DNA fragments amplified from genomic DNA extracted from the caudal fin. The amplification was performed using forward (5′- GGTGTTCAAG CTTATAGCTATGGACCCTTAC-3′) and backward (5′- TGTAGGGTATCTTCTGCAATGTTGCAG-3′) primers for *scn4aa*, and forward (5′- GACACGGAAAATGTGCGGAACTGGTG-3′) and backward (5′- CAACAGGATGCATATACCTACCAACTGG-3′) primers for *scn4ab*. These primers were designated *scn4aa*F, *scn4aa*R, *scn4ab*F, and *scn4ab*R, respectively.

### Video recording of zebrafish behavior

Escape responses were measured and analyzed as previously described [13]. Briefly, the trunk of an embryo was mechanically stimulated with a plastic pipette to induce escape behavior. In some experiments using 2% (w/v) methylcellulose solution, the tail of fish was pinched with forceps to elicit responses. Images were obtained using a Stemi 2000-C stereomicroscope (ZEISS) at 1,000 frames per second (fps) using a FastCaM 1,024PCI high-speed camera (Photron). The data were acquired as successive images. For measurement of the neck angle, a line was manually drawn from the swim bladder to the midpoint between eye balls for each image on ImageJ (version 1.52), whose angle was measured by the "measure" function. The angle values were normalized to the first image such that the angle of the first image became zero and were plotted against time. The turn speed (degrees/ms) was calculated as the change in the angle. When recorded in methylcellulose, the obtained plot was not smooth, so smoothing manipulation was performed using the savgol_filter() function from the scipy.signal library (version 1.11.3) in Python 3.10.12 before calculating the angle.

To record coiling activity, the chorion was manually removed, and embryos obtained at 28–32 hours post-fertilization (hpf) were embedded in low melting temperature agarose (NIPPON GENE). Coiling behavior was evoked by tactile stimuli on the dorsal side. Locomotion was recorded using a high-speed camera at 500 fps, and the shutter speed was 1/3000 s under the stereo microscope. The angle of the tail tip relative to the original position was measured using ImageJ. Turn speeds (degree/ms) were calculated as the change in the angle.

### RNA sequencing

Adult male zebrafish (two WT and one NaVDKO) were anesthetized, and the trunk muscles were dissected. Total RNA was extracted from the isolated tissue using ISOGEN (NIPPON GENE) following the manufacturer's instructions, including

digestion with DNase I. The RNA was sequenced in paired mode on an Illumina HiSeq6000 by Rhelixa, Inc. (Tokyo, Japan). The quality of the sequence reads was assessed using FastQC software (version 0.11.7). Low-quality bases and adaptor sequences were trimmed using the Trimmomatic software (version 0.38). The trimmed reads were aligned to the zebrafish genome *GRCz11* using HISAT2 (version 2.1.0).

### Ca$^{2+}$ imaging

The promoter region of *mylz2* (approximately 2 kb) was amplified from 5′-aaa-gaattc-actagt-ATTCGCCACAGAGGAATGAGCC-3′ and 5′-tttgtcgacGTGTGAAGTCTAAGAAGATCAAGAAGAGAAGTC-3′ primers and subcloned between Tol2 right and GCaMP7a sequence of the pT2RUASGCaMP7a plasmid (Kawakami lab). For in vivo Ca$^{2+}$ imaging, the generated pT2G-mylz2pro plasmid was injected into single-cell stage WT and NaVDKO embryos. Embryos were prepared as described previously [13]. Briefly, the fish were anesthetized using N-benzyl-p-toluene sulfonamide (BTS) [14] and mounted on low-temperature melting agarose. Spontaneous activity is induced by N-methyl-d-aspartate (NMDA) [15]. Images were acquired using an SP8 confocal microscope (Leica) equipped with a 5×/0.15 objective lens. Sixty-four pixel-square regions were scanned every 41 ms.

For Ca$^{2+}$ imaging of the isolated myocytes, the transgenic line Tg[mylz2-GCaMP7a] was generated by injecting pT2G-mylz2pro into single-cell stage WT embryos. Isolated myocytes were transferred to the 100 µL Tyrode's solution droplet on the coverslip. Subsequently Tyrode's solution containing 10 µM acetylcholine (ACh, 5 µL) was gently applied to stimulate myocytes. For the tetrodotoxin (TTX) assay, TTX was added to the droplet on the coverslip (final concentration, 1 µM). The ACh solution also contained 1 µM TTX. The myocytes were used for the assay within 6 h of preparation. Images were acquired using an EOS RP camera (Canon) attached to an MVX10 microscope (Olympus). For analysis, color images were converted to 8-bit monochrome images by ImageJ (Fiji, version 1.54f). The fluorescence intensity of each pixel was indicated in colors using the "fire" look-up table in ImageJ.

### Isolation of myocytes

Embryos or adult zebrafish were anesthetized using MS-222. The trunk was digested by 3 mg/mL type I collagenase (Sigma-Aldrich) in Tyrode's solution containing 110 mM NaCl, 2.1 mM KCl, 1.25 mM MgCl$_2$·6H$_2$O, 0.28 mM NaH$_2$PO$_4$, 3.5 mM NaHCO$_3$, 5 mM glucose, 15 µM CaCl$_2$, 10 mM HEPES, and 4 mM pyruvate; the pH was adjusted to 7.5. Myocytes were washed in Tyrode's solution and the CaCl$_2$ concentration was gradually increased to 1 mM [16].

### Electrophysiology

The sodium current in the isolated myocytes was recorded as previously described [17]. The pipette solution contained 58 mM CsCH$_3$SO$_3$, 32 mM CsCl$_2$, 10 mM EGTA, 10 mM HEPES (pH 7.1). The bath solution contained 100 mM NaCl, 2 mM KCl, 0.2 mM CaCl$_2$, 28 mM MgCl$_2$, 3 mM glucose, and 5 mM HEPES (pH 7.4). The leak currents and the current for charging the cell capacitance were subtracted using the P/-10 protocol. The sampling frequency was set at 100 kHz. The pulse interval was set at 5 s. "Gating" current of the DHPR was recorded in the presence of 1 µM TTX in the bath solution. The recording conditions, including the bath solution, pipette solution, pulse protocol, and holding potential, were identical to those used for the sodium current recording. To identify the genotype of isolated myocytes, genomic DNA was isolated from the head of the dissected fish myocytes. DNA fragments were amplified using the primers *scn4aa*F, *scn4aa*R and *scn4ab*F, *scn4ab*R, respectively (as detailed in the Genome Editing section), and subsequently sequenced. To record calcium currents, the pipette solution contained 100 mM CsCl$_2$, 20 mM CsF, 2 mM MgCl$_2$, 0.1 mM EGTA-Cs, 10 mM HEPES, 40 mM glucose, with the pH adjusted to 7.4 using CsOH. The bath solution contained 130 mM tetraethylammonium (TEA)-Cl, 10 mM CaCl$_2$, 1 mM MgCl$_2$, 20 mM glucose, 10 mM HEPES, with the pH adjusted to 7.4 using TEA-OH. The leak currents and the current for charging the cell capacitance were subtracted using the P/-10 protocol. The sampling frequency was set to 50 kHz. The pulse interval was set at 7 s. Recording of the miniature end-plate current

(mEPC) was recorded as described previously with some modifications [13,18]. Embryos were skinned and pinned in the recording chamber and spontaneous activity was recorded in a whole-cell configuration. Membrane potential was held at −90 mV. The bath solution contained 100 mM NaCl, 2 mM KCl, 0.2 mM CaCl$_2$, 2.8 mM MgCl$_2$·6H$_2$O, 3 mM glucose, and 5 mM HEPES (pH 7.4). Nifedipine (Merck, 10 µM) and TTX (1 µM) were added before recording. The pipette solution contained 120 mM KCl, 5 mM EGTA, and 5 mM HEPES (pH 7.2). The sampling frequency was set at 100 kHz. The series resistance was lower than 10 MΩ and compensated 65%–80%. Miniature events with amplitudes larger than 0.1 nA and coefficients of determination higher than 0.97, as determined by exponential fitting, were analyzed. Data were acquired using an EPC10 patch-clamp amplifier with PatchMaster software (version 2 × 73, HEKA Electronik) and analyzed using custom-made programs in Python 3.10.12.

## Simulation of muscle membrane potential

The simulation was performed using the NEURON software (https://www.neuron.yale.edu/neuron/) (version 8.2.3). The myofiber was modeled as a cylinder and the potential amplitude was calculated. ACh release sites were incorporated as cylinders 1 µm in length and 16 µm in diameter. The Hodgkin–Huxley model provided in the software was used under zero sodium conductance to mimic NaVDKO. The default values were used for the other parameters, except for the resistivities of the membrane, fiber interior, and diameter of the cylinder.

## Immunohistochemistry

Zebrafish embryos were anesthetized using MS-222 and fixed with 4% paraformaldehyde (PFA) for 4 h at room temperature. After rinsing with phosphate buffered saline including 0.1% Tween 20 (PBS-T) and dehydrated by washing in 100% methanol, samples were stored at −20 °C. For immunostaining, embryos were hydrated in a series of washes with 75%, 50%, and 25% methanol and PBS. The embryos were permeabilized with 1 mg/mL collagenase type I (Fujifilm Wako) in PBS for 3 h at room temperature. After rinsing with PBS-T, samples were incubated overnight at 4 °C with the anti-synaptic vesicle protein 2 (SV2) antibody (Developmental Studies Hybridoma Bank) at 1:200 dilution in blocking solution containing 5% sheep serum (Sigma-Aldrich) in PBS-T. After washing with PBS-T, samples were incubated overnight at 4 °C with goat anti-mouse IgG secondary antibody conjugated with Alexa 555 (Thermo Fisher Scientific) at 1:500 dilution and α-bungarotoxin (BTX) Alexa Fluor 488 conjugate (Molecular Probes) at 1:500 dilution in blocking solution. Images were acquired using an SP8 laser scanning confocal microscope (Leica) equipped with a 20 ×/0.75 objective lens in sequential mode.

For analysis, z-stack color images were superimposed and split into individual red, green, blue images using "split channels" in ImageJ (Fiji, version 1.54f). "R" and "G" images were adjusted using the "Threshold" tool to eliminate background signals. The areas stained with BTX and SV2 were estimated from the number of positive pixels and the matched fraction of BTX and SV2 signals was defined as follows: (number of positive pixels for both BTX and SV2)/ (number of positive pixels for either BTX or SV2). Positive pixels were counted using custom-made programs in Python 3.10.12.

For BTX staining of isolated myocytes, myocytes after dissociation were fixed in 4% PFA at room temperature for 30 min. After washing by PBS-T, myocytes were stained by BTX Alexa Fluor 488 conjugate (Molecular Probes; 1:400 with PBS-T) at room temperature for 1 h in the dark. Fluorescence was observed using a BZ- × 700 fluorescence microscope (KEYENCE).

## Swimming treadmill for adult fish

The 170 mL model SW10000 swim tunnel respirometer (Loligo Systems) was used. The water velocity was calibrated according to the manufacturer's instructions. The oximeter was calibrated using 0% and 100% O$_2$ saturation. Five percent (w/v) Na$_2$SO$_3$ and water aerated for 1 h were used for 0% and 100% O$_2$ solutions, respectively. A critical swimming speed (Ucrit) was calculated as (U + T)/ 60, where U is the actual flow rate at which fish terminated swimming and T is the time fish spent swimming at the flow rate [19,20]. Three-month-old fish were used for analysis. Water temperature ranged from 24.9–26.5 °C and the oxygen saturation exceeded 89%. In the Ucrit analysis, the regression line was estimated using the polyfit() function in the NumPy library version 1.26.0. in Python 3.10.12.

## Statistics

In the bar graphs, the mean and standard error of the mean (SEM) are shown, in addition to individual dots. *P*-values were calculated using the Welch's *t* test. Welch's *t* test and Pearson's correlation coefficient were calculated using the ttest_ind() function in the scipy.stats library version 1.11.3 and the corrcoef() function in the NumPy library version 1.26.0, respectively, in Python 3.10.12.

## Results

### Generation of *scn4aa* and *scn4ab* double knockout zebrafish

We generated *scn4aa* and *scn4ab* heterogeneous zebrafish by the CRISPER-Cas9 method. Together, these two genes encode NaV1.4. Because of genome duplication during evolution, it is common for two genes in zebrafish to correspond to a single human gene, in this case, *SCN4A* (S1 Table). Four bases were deleted in exon 14 of *scn4aa* gene and 14 bases were inserted in exon 8 of the *scn4ab* gene (S1A1 and S1B1 Fig). These changes cause frameshifts and create premature stop codons (S1A2 and S1B2 Fig). Premature stop codons were located between S3 and S4 of domain II for *scn4a*a, and between S5 and S6 of domain I for *scn4ab* (S1C Fig). According to structural studies of NaVs [21,22], the remnant protein is expected to be nonfunctional even if it reaches the plasma membrane.

Subsequent crossing of the two lines generated a double-heterogeneous fish that was identified by sequencing. The *scn4aa*/*scn4ab* double knockout line (NaVDKO) was generated by crossing two heterogeneous fish. To verify the functional abolition of NaV1.4 proteins in muscles, we recorded the voltage-dependent sodium current from isolated trunk myofibers of embryos. Trunk muscle comprises two types of myocytes: superficial and deep muscles (i.e., slow and fast fibers). In WT animals, slow fibers do not have voltage-dependent sodium currents [23]. As positional information is lost after myocyte dissociation, we recorded currents without distinguishing between fast and slow fibers.

Current recordings from WT fibers in the presence of cesium in the pipette solution showed transient inward currents following the start and end of depolarizing pulses in 24 cells from nine fish (upper panel in Fig 1A). The inward current in the initial phase (<5 ms) of the depolarization pulse disappeared by the application of 1 µM TTX (middle panel in Fig 1A), demonstrating that the current was derived from NaVs [14]. The residual ON- and OFF-current recorded in the presence TTX is therefore the "gating" current of the DHPR, that is, voltage dependent calcium channel (middle panel in Fig 1A) [24].

When we recorded currents from 24 cells in seven NaVDKO fish, we observed only "gating" currents of the DHPR (bottom panel in Fig 1A and 1B). These results indicate that NaVDKO fish lack functional expression of NaVs in their myocytes.

*Caenorhabditis elegans* does not have a gene encoding NaV in its genome. The roles of NaV are substituted by $Ca^{2+}$-dependent spikes in excitable cells [25–27]. The previously described absence of calcium currents in zebrafish myocytes [17,28] was confirmed in the present study (Fig 1C and 1D). The findings indicate that NaVDKO embryos do not contain sodium APs or calcium spikes.

### Locomotion of embryos

Slow muscles which lack NaV expression [23] underlie the tail coiling movement observed in 1-dpf zebrafish embryos [29]. We examined the coiling activity of NaVDKO zebrafish to determine its potential effects on slow muscles. Rostral parts of 28–32-hpf embryos were embedded in a gel and coiling behaviors were recorded (S2A Fig). The deviation of the tail in the images was measured as the angle (S2B Fig). The maximum turn speeds of individual embryos were indistinguishable between WT and NaVDKO zebrafish (S2C Fig).

As embryos develop, rapid locomotion driven by fast muscles becomes dominant. We expected that NaVDKO zebrafish would have compromised quick locomotion owing to dysfunctional fast fibers, as observed in mutants whose fast muscles do not receive inputs from motor neurons [13]. The touch responses of the embryos were recorded. Normal embryos

 

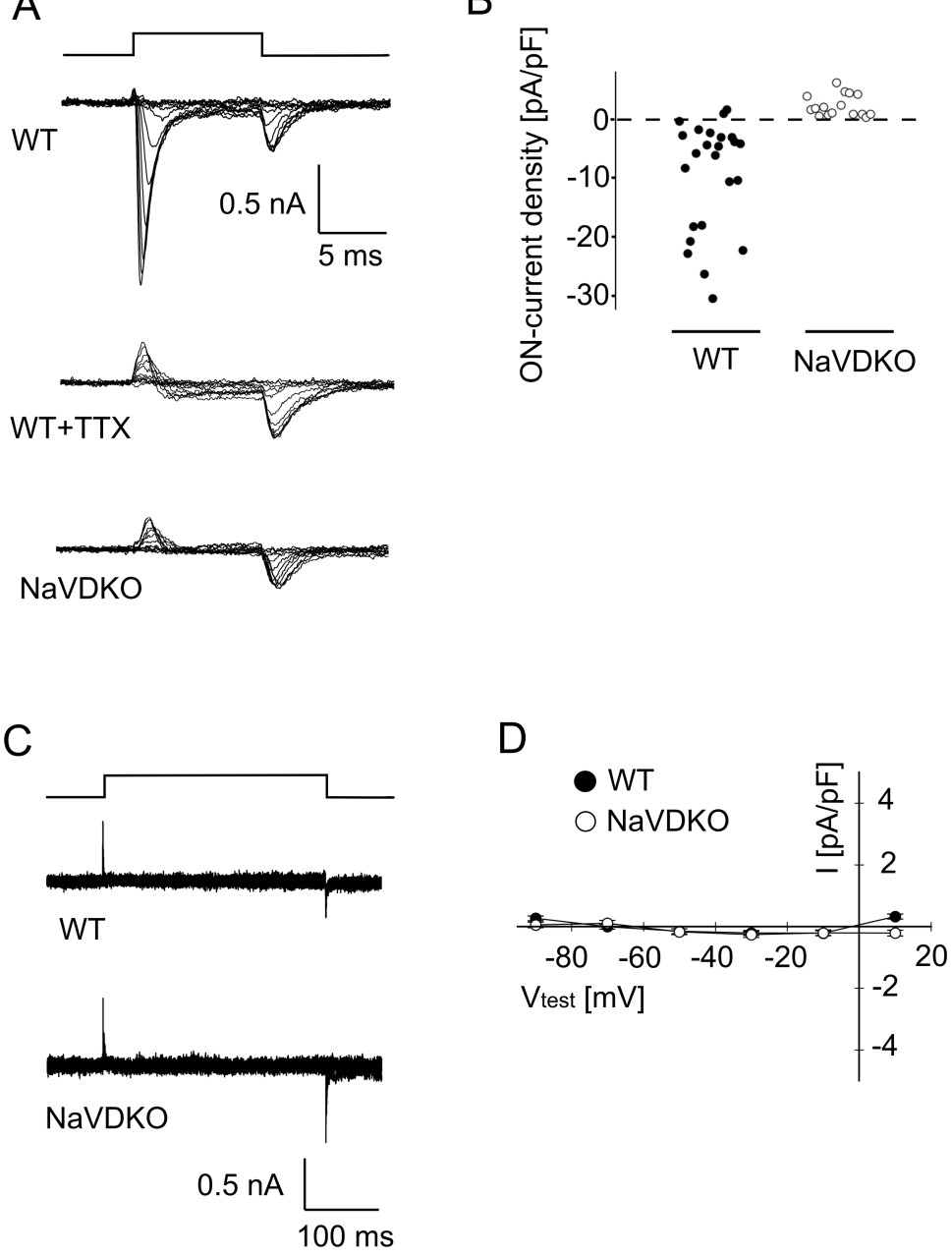

**Fig 1. Electrophysiology of isolated myocytes. (A)** Representative sodium current of WT (upper panel), WT in the presence of 1 μM TTX (middle panel) and NaVDKO (lower panel) myocytes. Top panel indicates the timing of the depolarization pulse. Traces evoked by voltages ranging from −90 to 20 mV in 10 mV increments are superimposed. **(B)** Comparison of the peak ON-current density between WT and NaVDKO myocytes. Currents were recorded from myocytes at 3–6 days post fertilization (dpf). **(C)** Representative calcium current recordings from isolated myocytes. Top panel indicates the timing of the test pulse. Traces evoked by voltages ranging from −90 mV to 10 mV in 20 mV increments are superimposed. **(D)** Current–voltage relationship of the calcium current recording. Average current amplitudes 100–102 ms after the onset of the test pulse were plotted against potentials. Data were shown as mean±SEM (n=6 for both WT and NaVDKO). The numerical data presented in this figure can be found in S1 Data.

display fast and large body bends (C bends), followed by smaller counter bends [30,31]. Unexpectedly, NaVDKO embryos exhibited robust escape responses comparable to those of WT embryos (Fig 2A and 2B and S1 and S2 Movies). The maximum turn angle of the C-bend (the first turn) and the maximum turn speed were not statistically different from those of the WT (Fig 2C and 2D).

Considering that differences can be observed under exigent conditions, we tested the escape response of embryos in a viscous solution containing 2% methylcellulose, in which embryos required more power for escape. After pinching the tail, the embryos displayed smaller turn angles than those in water (S3A, S3B and S3C Fig and S3 and S4 Movies). The maximum angle of the turn calculated from the smoothed curve (S3B Fig) was again indistinguishable between WT and NaVDKO (S3C Fig). The maximum turn speed of the NaVDKO was comparable to that of the WT zebrafish (S3D Fig). The collective findings indicated no significant difference in the escape response between WT and NaVDKO zebrafish.

**Lack of compensatory mechanisms in NaVDKO**

The normal locomotion of NaVDKO (Figs 2, S2 and S3) prompted us to examine whether the upregulation of other NaVs compensated for the deleted NaVs. RNA sequencing of trunk muscles from WT and NaVDKO zebrafish (S2 and S3 Tables). In the analysis of differentially expressed genes (DEG), among eight *scn* genes of zebrafish, only *scn8aa* and

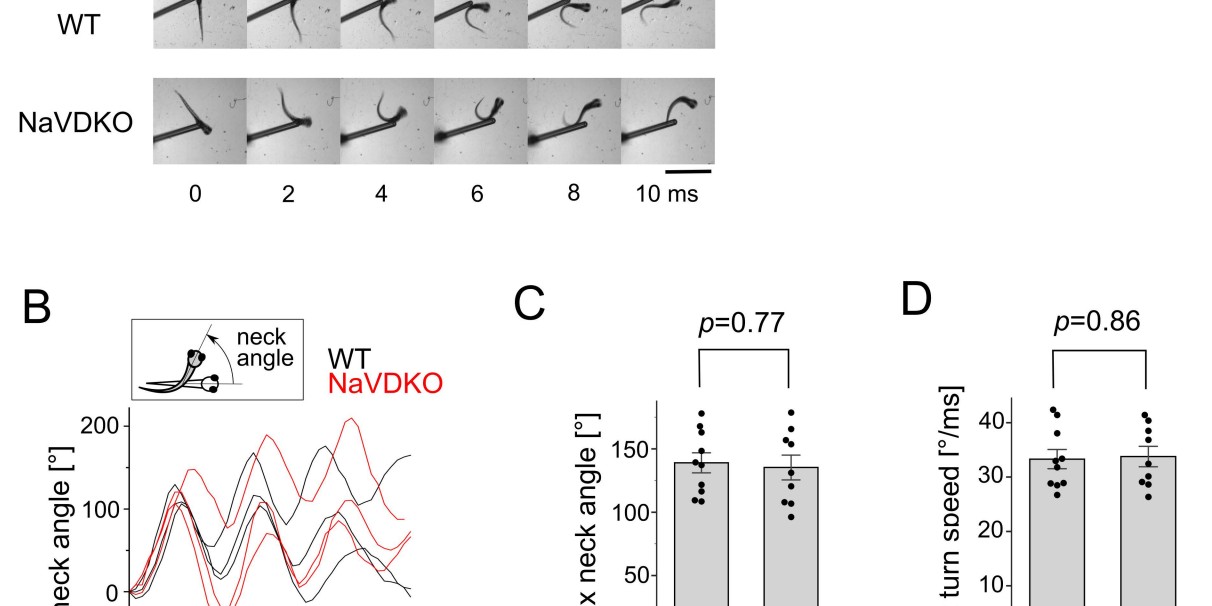

**Fig 2. Escape response of WT and NaVDKO zebrafish. (A)** Snapshots of the representative escape responses of WT (upper) and NaVDKO (lower) zebrafish at 4 dpf in normal water. Images were captured every 2 ms. Scale bar: 1 mm. **(B)** Representative plots of the "neck angle" during the escape behavior of WT and NaVDKO fish. The inset shows the definition of "neck angle." **(C)** Analysis of the maximum neck angle in the first turn of 4 dpf fish. Individual data points are shown as dots. Bars and error bars represent the mean and SEM, respectively (n = 10 for WT, n = 9 for NaVDKO). **(D)** Analysis of the maximum turn angle speed of 4 dpf fish. The maximum turn angle speed was calculated from the differential of the neck angle (n = 10 for WT, n = 9 for NaVDKO). The numerical data presented in this figure can be found in S1 Data.

*scn12aa* showed positive logFC values (0.173 and 0.158 for *scn8aa* and *scn12aa*, respectively; S4A Fig and S4 Table). However, the respective *p*-values of 0.881 and 0.935 for *scn8aa* and *scn12aa* did not reach level of significance (S4 Table). Moreover, the transcript per million (TPM) values for *scn8aa* and *scn12aa*, were negligible in both the WT and NaVDKO embryos compared to those for *scn4aa* and *scn4ab* (S4 Table and S4B Fig). The lack of robust upregulation among *scn* genes in NaVDKOs is compatible with the absence of Na$^+$ currents in the myocytes of NaVDKOs (Fig 1A and 1B).

We also examined alpha1, a pore-forming subunit of calcium channels. The only gene that displayed significant upregulation was *cacna1ia*, which encodes CaV3.3 (S5 and S6 Table). We examined whether upregulation of *cacna1ia* leads to the appearance of calcium currents in myocytes. Calcium currents were not recorded in myocytes of NaVDKO or WT zebrafish (Fig 1C and 1D). The negligible level of TPM of the *cacna1ia* was consistent with the current recordings. In summary, the sodium or calcium channels in NaVDKO do not compensate for the loss of NaV1.4.

We also examined the possibility that the deletion of NaV1.4 is compensated by remodeling of the NMJ. Pre-and post-synapses were stained with antibodies against SV2 and alpha-BTX conjugated with Alexa488, respectively. At 28 hpf, nicotinic ACh receptors (nAChRs) and SV2 were observed in the middle body segments (arrowhead in S5A$_1$ Fig) and spinal cord (arrow in S5A$_1$ Fig) [32]. Areas with signals above the thresholds (S5A$_2$ Fig) were measured. Areas of positive signals were not different between WT and NaVDKO embryos (S5A$_3$ Fig), and the double-positive fraction for BTX and SV2 signals was not distinct between the two (S5A$_4$ Fig). In 3 dpf embryos, BTX and SV2 signals were observed in both types of NMJs. Signals at the boundaries between body segments belong to slow muscles (arrow in S5B$_1$ Fig) and punctate signals in body segments were NMJs of fast muscles (arrow head in S5B$_1$ Fig) [12,33]. The signal-positive area and double-positive fraction did not differ between WT and NaVDKO zebrafish (S5B$_2$, S5B$_3$ and S5B$_4$ Fig). In 5 dpf embryos, the number of NMJs increased compared with 3 dpf embryos (S5B$_1$ and S5C$_1$ Fig), whereas no difference was observed between WT and NaVDKO embryos (S5C$_2$, S5C$_3$ and S5C$_4$ Fig). These data suggest that the knockout of NaV1.4 is not compensated by the NMJ remodeling.

## Ca$^{2+}$ imaging of muscle fibers

To explore NaVDKO embryos at the cellular level, we examined changes in intracellular calcium ion concentration ([Ca$^{2+}$]$_i$) in the trunk muscles of WT and NaVDKO embryos. [Ca$^{2+}$]$_i$ was visualized by transient expression of the genetically encoded Ca$^{2+}$ indicator GCaMP [34]. GCaMP expression was driven by the fast fiber-specific *mylz2* promoter (S6A Fig) [35]. Notably, GCaMP was ectopically expressed in some slow fibers that could be anatomically distinguished from fast fibers. The embryo was anesthetized and spontaneous activity was induced by the application of NMDA. We found a transient increase in [Ca$^{2+}$]$_i$ in the WT (S6B Fig, upper panels and S5 Movie). These events concurrently occurred in fast and slow fibers (S6C Fig). Increase in [Ca$^{2+}$]$_i$ was also observed in NaVDKO embryos, which was not distinct from that in WT embryos (S6B and S6D Fig and S6 Movie).

The results presented thus far indicate that NaVs are dispensable for muscle contraction. If this is the case, WT muscle fibers are expected to contract even in the presence of TTX. To ensure penetration of TTX into fast muscle fibers, myofibers were dissociated using collagenase. To identify fast fibers, transgenic zebrafish expressing GCaMP7a driven by the *mylz2* promoter were generated.

First, we stimulated myofibers by 10 μM ACh in the absence of TTX. For S7A$_1$ Fig, ACh was applied along the indicated directions. The timing of the fluorescence change varied depending on the location of the cell relative to the ACh application (Figs 3B, S7A$_1$ and S7A$_2$ and S7 Movie).

Next, Ca$^{2+}$ imaging was performed in the presence of 1 μM TTX. The application of ACh and TTX induced a robust increase in fluorescence (Figs 3C and S7B and S8 Movie). NaV currents were completely blocked by 1 μM TTX (Fig 1A, upper and middle panels), these data again supported the notion that NaVs are dispensable for the myofiber contraction in zebrafish embryos.

## Swimming capability of adult NaVDKO

After concluding that NaVs are dispensable for embryonic locomotion, we hypothesized that NaVs play an essential role in adults. Accordingly, we assessed the swimming capability of the adult fish. Escape behavior was evoked by pinching the tail in methylcellulose in the same manner as in the embryos. The locomotion of adult NaVDKO and WT fish was compared (S8A and S8B Fig and S9 and S10 Movies), and no significant difference was found in the maximum angles of the first turn (S8C Fig). The maximum turning speed calculated from the smoothed curves did not differ (S8D Fig). The findings do not indicate compromised locomotion in adult NaVDKO zebrafish.

To examine another aspect of locomotion, adult NaVDKO zebrafish were examined using a swimming treadmill. In the swim tunnel, adult fish swim in a cylindrical tube in which the flow rate is controlled by the rotation speed of the screw located at the end of the cylinder (S9A Fig). The flow rate was maintained at 15 cm/s for the first 1 min as the fish acclimated to the tube. Measurement of the Ucrit started at 20 cm/s. The flow rate was increased by 1 cm/s every minute until the fish stopped swimming (S9B Fig).

While a weak negative correlation was found between Ucrit and body mass index (BMI), Ucrit was not associated with the fish genotype (S9C and S9D Fig). Therefore, the swimming ability of adult NaVDKO was comparable to that of WT fish.

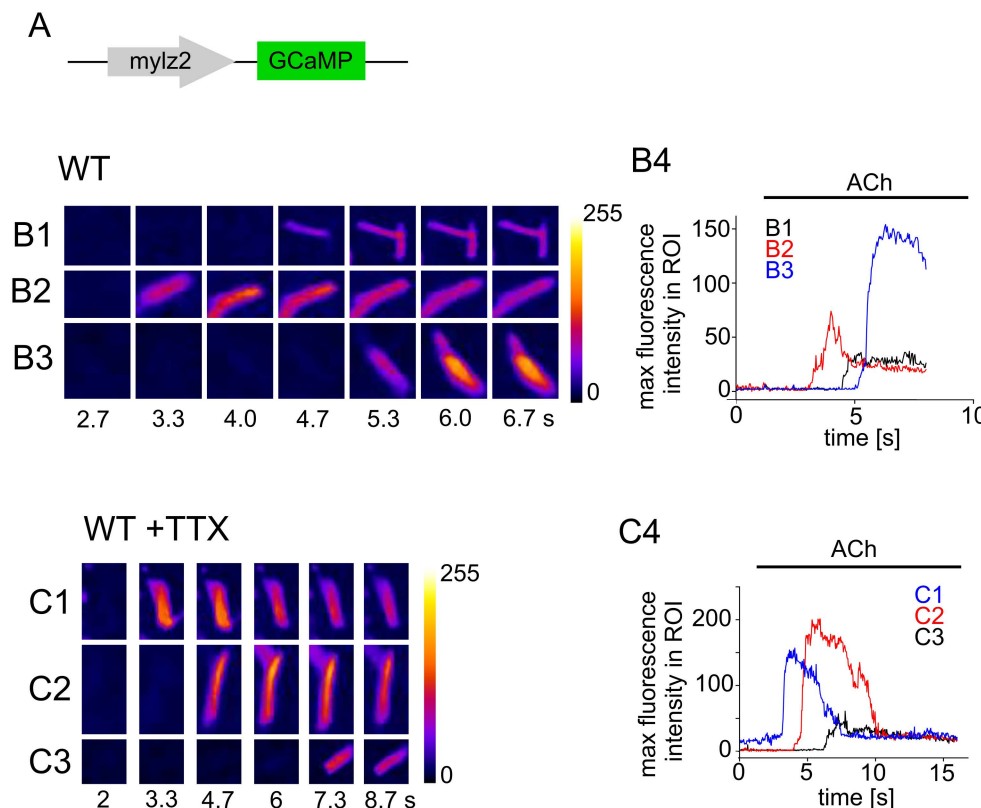

**Fig 3. Ca²⁺ imaging of isolated myocytes in the absence and presence of TTX. (A)** Expression of GCaMP7a was driven by the *mylz2* promoter. (B1, B2, and B3) Representative Ca²⁺ imaging of WT myocytes at 3 dpf embryos. Numbers below the images indicate time from the beginning of the recording. Fluorescent intensity is shown in color. (B4) Maximum fluorescence intensity in the region of interest (ROI) (B1, B2 and B3) was plotted against time. (C1, C2, and C3) Ca²⁺ imaging of WT myocytes at 5 dpf in the presence of 1 μM TTX. (C4) Maximum fluorescence intensity of ROI shown in C1, C2 and C3 were plotted against time. The numerical data presented in this figure can be found in S1 Data.

## Simulation of muscle excitation without voltage-dependent sodium conductance

The EPP activates NaVs, and the generated AP spreads by activating neighboring NaVs until depolarization reaches the DHPR in the T-tubule. The amplitude of the EPP is generally smaller than that of the AP, which is insufficient for the stimulation of DHPR [36]. Therefore, it was unexpected that the NaVDKO zebrafish would show normal locomotion.

DHPRs in zebrafish myofibers lack $Ca^{2+}$ conductance (Fig 1C and 1D) [17,28]. Moreover nAChRs in fast muscles has a low $Ca^{2+}$ permeability [13,37]. Therefore, the sole source of $Ca^{2+}$ for contraction is the SR, for which DHPR activation is crucial. One possible mechanism for the contraction of NaVDKO fibers is the direct activation of DHPRs by EPPs.

To examine whether sufficiently large depolarization can be transmitted along the plasma membrane without NaVs, we performed simulations using NEURON software. The muscle fiber was modeled as a cylinder for simplicity and ease of computation (Fig 4A); a short cylindrical section was inserted as neuromuscular synapses, and the synaptic currents were injected in this area (dark gray cylinder in Fig 4A). Depolarization was evoked by current injection at the ACh release site and was conducted in the light gray section, where the time-dependent change in the potential was calculated. According to the cable theory [38], the parameters of membrane resistivity and fiber interior are key factors for attenuation. However, to the best of our knowledge, the resistivity of zebrafish myofibers has not been reported. Therefore, we used an extensor longus digit. IV for the European frog [39]. The parameters used in the simulation are summarized in S7 Table.

We initially employed the one-synapse model shown in Fig 4A to focus on the attenuation of depolarization along the fiber length in the absence of NaV. Potentials at positions 0.1, 0.5, and 0.9L from the ACh release site (L = length of the cylinder) in the light gray cylinder (Fig 4A) were calculated when the current was injected at the ACh release site. The current amplitude and duration were set as 15.6 nA and 0.6 s, respectively. When the fiber length was 100 μm (L = 100 μm), the maximum amplitude of depolarization was around 80 mV, which was comparable between 0.1 and 0.9L (Fig 4B$_1$), indicating the negligible attenuation of the potential. When the fiber length was changed from 100 to 200 μm, the membrane depolarization was reduced to approximately 0 mV, whereas attenuation was not significant (Fig 4B$_2$). Smaller depolarization and noticeable attenuation were evident when the length was set to 500 or 1,000 μm (S10A and S10B Fig).

Muscle fibers in zebrafish embryos were 83.1 ± 5.3 and 101.0 ± 5.8 μm in length for WT and NaVDKO, respectively (data are expressed as mean ± SEM, with 25 cells for WT and 28 cells for NaVDKO) (Fig 4C$_1$ and 4C$_2$). The findings indicate that attenuation of the membrane potential is not significant in fibers at the embryonic stages.

Next, we simulated the change in membrane voltage in a fiber with a current amplitude and synapse distribution closer to the biological situation. A previous report showed that the mean quantal content of the neuromuscular junction (all ACh release sites) in zebrafish embryos is approximately 12 [18]. Quantal content is defined as the number of presynaptic vesicles released into the synaptic cleft(s) of a single motor neuron by the AP. We counted the number of ACh release sites per fiber by BTX staining; 13.1 ± 0.9 for WT (n = 27) and 12.4 ± 0.9 for NaVDKO (n = 19) (Fig 4C$_1$ and 4C$_2$). These numbers are close to the quantal content, which may indicate that one synaptic vesicle was released at a single ACh release site.

We recorded the mEPC (Fig 4D). Mean amplitude was estimated by fitting the histogram plot using Gaussian distribution (Fig 4E). The mean amplitude value was 1.2 ± 0.1 nA for WT and 1.3 ± 0.2 nA for NaVDKO (n = 12 for both). The decay constant of mEPC obtained by fitting to the exponential equation was 0.58 ± 0.03 ms for WT (n = 4,538 events) and 0.61 ± 0.04 ms for NaVDKO (n = 6,127 events). To simulate currents with these parameters, the synaptic input was modeled as a square pulse with the amplitude and the width set to 1.2 nA and 0.6 ms at individual release sites. ACh release sites were diffusely disseminated in fast muscle fibers (Fig 4C). For simulation, 13 release sites were inserted at equal intervals (Fig 4F). The myofiber cylinder was separated into 12 regions (1–12), where the membrane potential was calculated. Assuming that the release probability of all ACh release sites is one, which means that a 1.2 nA current was simultaneously injected at all sites, the maximum depolarization in the middle of individual regions was approximately 80 mV (Figs 4G and S11A). Recordings of the "gating" current from isolated myofibers revealed that the DHPR was fully activated when the membrane was depolarized at approximately 80 mV (S11B Fig), suggesting that DHPRs can be fully activated by the EPP alone.

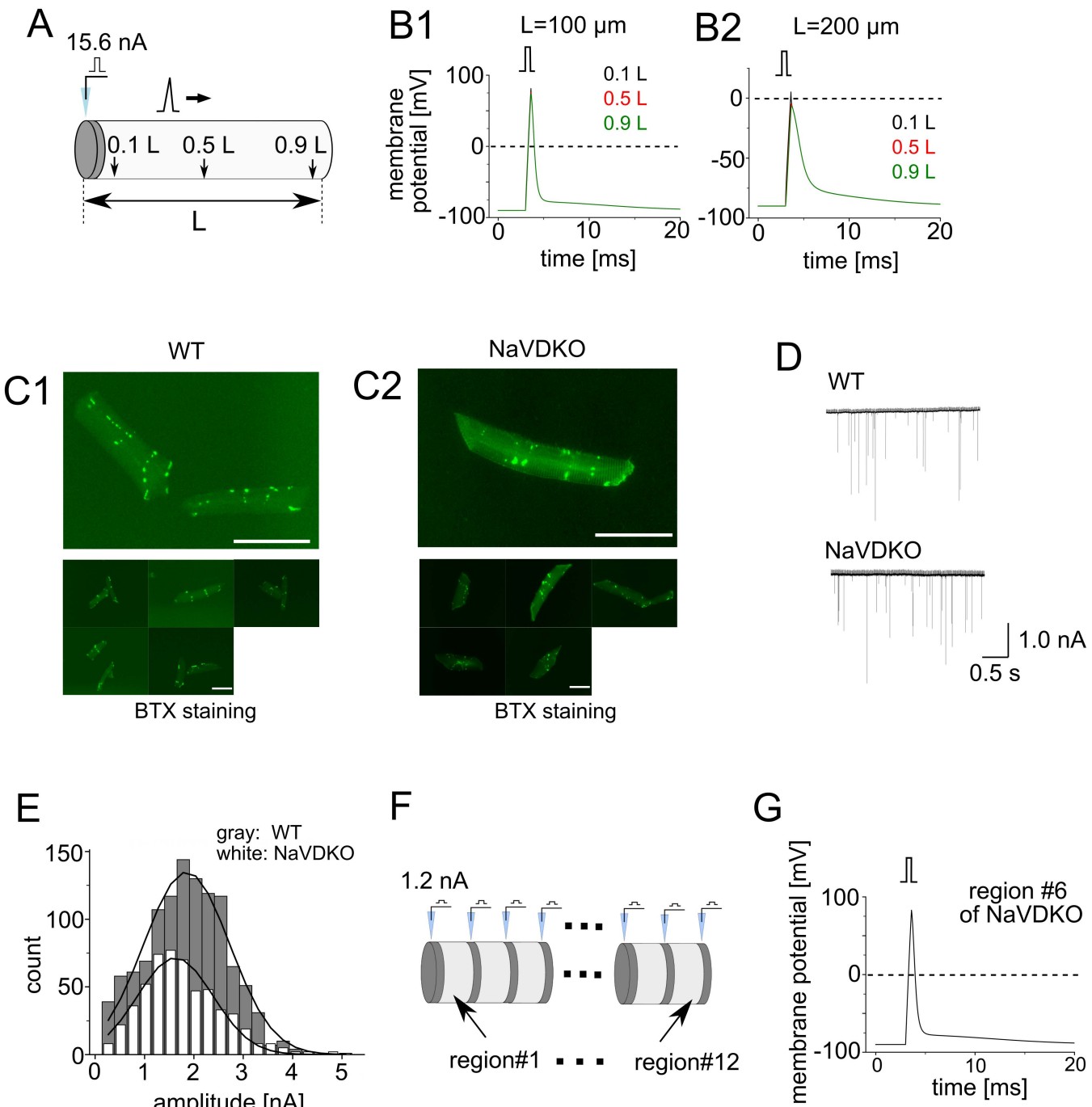

Fig 4. Mathematical simulation of depolarization in embryonic myocytes. (A) A single synapse model with a length of **L.** The ACh release site was modeled as a 1 μm-long cylinder (dark gray). The current was injected at the ACh release site and potentials at 0.1, 0.5, and 0.9L away from the ACh release site were calculated. (B) Membrane potentials for L = 100 (B1) and 200 μm (B2). Potentials at 0.1, 0.5, and 0.9L from the ACh release site were calculated. Three colored traces (0.1, 0.5, and 0.9L) overlapped. Square pulses above the plots indicate the timing of the synaptic input. (C) BTX staining of myofibers in 4 and 6 dpf fish. Myofibers were from WT (C1) and NaVDKO (C2) fish. Upper panels show magnified views. Scale bars: 50 μm. (D) Representative miniature end-plate currents (mEPC) of WT (left) and NaVDKO (right). (E) Representative histogram of the mEPC amplitude in WT (dark gray) and NaVDKO (light gray). The curves were fit to a Gaussian function. (F) Geometry of the multiple-synapse model. Thirteen 1 μm-long cylinders (dark gray) and 12 7 μm-long cylinders (light gray) were placed alternately. Synaptic currents were injected into individual ACh release sites. The 7 μm-long cylinders are designated as region #1 through #12. (G) Potentials at the center of region #6. The numerical data presented in this figure can be found in S1 Data.

To examine the dependency of the obtained results on the employed parameters, we performed a simulation using the resistivities of axons in squid giant axon [40], lobster leg axon [38] and goldfish Mauthner axon in goldfish [41] (S7 Table). In the geometry shown in Fig 4F, robust depolarization exceeding +30 mV was observed for all parameters (S12 Fig). These estimates suggest that DHPR is fully activated even in the absence of NaVs in the myofibers of embryos.

Our analyses revealed the normal locomotion of adult NaVDKO fish (S8 and S9 Figs). The adult myofibers were much longer than those of the embryos (1,102±87 μm, mean±SEM, n=16; S13A Fig). To examine whether EPP can directly stimulate DHPR in adults, we employed another set of parameters to simulate adult muscle fibers. The resistivities and amplitude of the synaptic current did not change in the embryos because the current could not be recorded by conventional patch-clamp techniques owing to the size of adult fibers.

We could not determine the number of ACh release sites per fiber because some were linear rather than punctate (S13A Fig). To reconcile this observation with the simulation, a range of ACh release site numbers per fiber were applied: 11, 51, 101, and 139 (S13B Fig). The potentials in the middle of this region are shown in S13C Fig. At sites 11 and 51, the maximum potentials were below 0 mV, whereas for the 101- and 139-site models, the maximum potentials were greater than 0 mV. The results of the 101- and 139-site models suggest that myofibers may be able to contract without NaVs, in which case the disseminated distribution of ACh release sites is the reason why NaVs are dispensable in adult fibers.

## Discussion

It is generally believed that APs generated by NaV1.4 in skeletal muscles are essential for muscle contraction. However, we found that *scn4aa*/*scn4ab* double knockout zebrafish (NaVDKO) exhibited uncompromised locomotion (Figs 2, S2, S3, S8 and S9). In agreement with this observation, $[Ca^{2+}]_i$ of WT myocytes was elevated, even in the presence of TTX (Fig 3C). Simulation using NEURON software revealed that the unattenuated amplitude of the EPP was sufficiently large for the full activation of DHPR in embryonic fibers (Figs 4G, S11 and S12). The same may be true for adult fibers if the number of synapses is sufficiently large (S13 Fig). Therefore, the direct activation of DHPR may enable muscle contractions without NaVs.

### Excitation of fast fibers without NaV1.4s

Contrary to the established view, this study showed that NaVs are dispensable for muscle contraction in zebrafish embryos and adults. Skeletal muscle fibers are roughly divided into fast and slow fibers, and NaVs are expressed only in fast fibers [23]. We previously knocked out the gamma and epsilon subunits of nAChRs specifically expressed in fast fibers, generating zebrafish lacking functional fast fibers [13]. These embryos were observed to swim much slower than WT fish. In sharp contrast, the swimming speed of the NaVDKO fish examined in this study was comparable to that of the WT fish (Figs 2, S2, S3, S8 and S9). These findings support the idea that fast fibers function normally in NaVDKO fish.

Ten genes encoding NaVs exist in human chromosomes, while the zebrafish genome contains eight α-subunit genes according to the genome assembly, GRCz11 (S1 Table). In situ hybridization of α-subunit genes in zebrafish embryos showed that *scn4aa* was expressed in the head, pharyngeal muscle, and pectoral fin at 72 hpf, whereas *scn4ab* was expressed in the trunk somites at 24 and 60 hpf [42]. Current recordings from isolated myofibers (Fig 1A and 1B) confirmed that NaV1.4, likely *scn4ab*, underlies the dominant sodium conductance in the myofibers of zebrafish.

### Physiological roles of NaV1.4s in zebrafish

Mutations in human NaV1.4, which alter skeletal muscle excitability, cause myotonia, periodic paralysis, congenital myopathy, and myasthenic syndrome. However, knockout of NaV1.4 did not affect zebrafish locomotion. Synaptic connections in mammalian skeletal muscles are located at the center of the myocytes (the innervation zone). In addition, mammalian myofibers are generally longer than zebrafish myofibers, extending more than a few centimeters. As shown in S10B Fig, the membrane depolarization of a 1 mm-long fiber was attenuated in the absence of sodium conduction. Therefore, the

distribution of ACh release sites and length of the fiber would be determinant factors for the physiological significance of NaV1.4.

APs have been reported in muscles of more primitive organisms than in fish. The amphioxus genome contains five genes encoding NaVs and the muscle contains sodium-dependent APs [43]. Muscle fibres in the ascidian *Halocynthia aurantium* have Ca$^{2+}$-dependent AP [44]. Five NaV1s and one NaV2 that are permeable to both sodium and calcium are found in the genome of the lamprey [45,46], among which NaV1γ expresses in muscles [47]. Despite the conservation of APs in the process of evolution from common ancestors of chordates, APs may not be necessary for muscle contractions per se in these organisms because the simulation performed in the present study suggested that the potential was not attenuated in short fibers. AP may be a redundant mechanism for depolarizing the membrane, which is sufficient for the full activation of DHPR. One possibility is that animals can save energy by using APs for muscle contraction. In zebrafish, individual muscle fibers receive input from a single primary motor neuron and up to three secondary motor neurons [48]. Because NaVs amplify EPP, the excitation of a single motor neuron may be sufficient for the excitation of myofibers. In contrast, excitation of all motor neurons may be necessary in the absence of APs. The present demonstration of the dispensability of NaV in the zebrafish muscle fibres sheds new light on the evolutionary roles of APs.

Although NaV2 has been lost in the zebrafish genome, NaV1.4s remains functional throughout evolution without becoming a pseudogene. This strongly suggests that NaV plays an essential role in zebrafish survival.

NaV channels are expressed in non-excitable cells such as immune cells, cancer cells, and osteoblasts in mammals [49]. The *scn8aa* gene in zebrafish encodes NaV1.6 that is expressed in motor neurons [50]. NaV1.6 is involved in the locomotor activity, development, innervation, and regeneration of fins [51–53]. In some electric fish, *scn4ab*1, a variant of *scn4ab*, is expressed in the electric organs [54]. In the sea urchin *Strongylocentrotus purpuratus*, NaV2 is expressed in the ventrolateral ectoderm during early developmental stages and plays a role in normal skeletal patterning [55]. The sea anemone *Nematostella vectensis* expresses NaV1 and NaV2 [56,57], which are not expressed in muscle fibres [57]. NaV1.4s in zebrafish may play a physiological role outside the skeletal system.

## Supporting information

**S1 Fig. Generation of *scn4aa/scn4ab* double knockout (KO) zebrafish.** (A1) Genomic sequences of the WT (left) and KO (right) zebrafish around the CRISPR target of *scn4aa*. The four bases in the dotted box were deleted from the mutant allele. (A2) Encoded amino acid sequences of the WT and KO zebrafish. Deletion of four bases highlighted in yellow engendered a stop codon (asterisk) in KO fish. (B1) Genome sequences of the WT (left) and KO (right) fish around the CRISPR target of *scn4ab*. The 14 bases in the dotted box were inserted into the mutant allele. (B2) Encoded amino acid sequences of the WT and KO fish. The insertion of the 14 bases highlighted in yellow generated a stop codon (asterisk) in the KO zebrafish. (C) Diagram of NaV with domains I–IV, each containing six transmembrane regions S1–S6. Positions of the mutations are indicated.
(TIFF)

**S2 Fig. Coiling activity of 1-dpf embryo.** (A) Representative coiling activities of WT and NaVDKO zebrafish. The images were superimposed every 2 ms. Scale bar: 0.5 mm. (B) The angle θ in images was measured as indicated. (C) Plot of maximum turn speed. The numerical data presented in this figure can be found in S1 Data.
(TIFF)

**S3 Fig. Escape response of embryos in methylcellulose.** (A) Representative escape responses of WT (upper) and NaVDKO (lower) embryos at 4 dpf in 2% (w/v) methylcellulose. The time elapsed since the start of recording is indicated below the image. Scale bar: 1 mm. (B) Representative plots of the "neck angle" during the escape behavior of WT and NaVDKO fish. (C) Analysis of the maximum neck angle in the first turn of 4 dpf fish (n = 10 for WT, n = 9 for NaVDKO). (D)

Analysis of the maximum turn angle speed of 4 dpf fish, calculated from the differential of the neck angle (n = 10 for WT, n = 9 for NaVDKO). The numerical data presented in this figure can be found in S1 Data.
(TIFF)

**S4 Fig. Differentially expressed genes (DEGs) analysis between WT and NaVDKO embryos.** (A) Volcano plot showing the $p$ values of DEGs. Red and blue dots indicate genes with $p$-values smaller than 0.05. Genes with log(FC) values larger than one and smaller than minus one are shown in red and blue, respectively. The numerical data presented in this figure can be found in S3 Table. (B) Transcripts per million (TPM) values for *scn8aa*, *scn12aa*, and *cacna1ia*. The numerical data presented in this figure can be found in S4 and S6 Tables.
(TIFF)

**S5 Fig. NMJ staining of WT and NaVDKO fish.** (A1, B1, and C1) BTX and anti-SV2 signals at 28 hpf, 3 dpf, and 5 dpf embryo, respectively. The BTX (left), anti-SV2 (middle), and merged (right) images are shown. The arrow and arrowhead in A1 indicate signals in the spinal cord and middle body segments, respectively. The arrow and arrowhead in B1 indicate signals at the boundaries between body segments and punctate signals in body segments, which are the NMJs of fast muscles. Scale bar: 50 μm. (A2, B2, and C2) Representative images showing pixels above the thresholds obtained from A1, B1, and C1. (A3, B3, and C3) Areas positive for BTX and SV2 in WT and NaVDKO embryos. (A4, B4, and C4) Overlap fractions of the BTX- and SV2-positive pixels. The numerical data presented in this figure can be found in S1 Data.
(TIFF)

**S6 Fig. In vivo Ca²⁺ imaging of trunk muscles using transient GCaMP expression.** (A) GCaMP7a expression was driven by the *mylz2* promoter. (B) Spontaneous muscle excitation in WT (upper) and NaVDKO (lower) fish. The numbers below indicate the time elapsed from the start of the recording. The fluorescence intensity is indicated by color. (C and D) Activities of WT (C) and NaVDKO fish (D). "A" and "P" in the upper panel indicate anterior and posterior, respectively. The mean fluorescence intensity for the pixels in the white boxes is plotted against time in the lower panels. F and S indicate fast and slow fibers, respectively. Fast and slow fibers were identified based on their orientation along the anterior–posterior axis of the body. The numerical data presented in this figure can be found in S1 Data.
(TIFF)

**S7 Fig. Ca²⁺ imaging of isolated myocytes in the presence and absence of TTX.** ($A_1$ and $B_1$) Snapshots of field images of the representative Ca²⁺ imaging of WT in the absence and presence of 1 μM TTX, respectively. The numbers indicate the time from the beginning of the recording. Fluorescent intensity is shown in color. White arrows in the first image indicate the direction of the puff application of Ach. Scale bar: 100 μm. (A2 and B2) Images of the myocytes analyzed in Fig 3. $B_1$, $B_2$, $B_3$, $C_1$, $C_2$ and $C_3$ correspond to the fibers from B1 to C3, respectively. Scale bar: 100 μm.
(TIFF)

**S8 Fig. Escape response of adult fish in methylcellulose.** (A) Representative escape responses of adult WT (upper panels) and NaVDKO (lower panel) fish. Images were captured every 10 ms. Scale bar: 1 cm. (B) Representative plots of the "neck angle" during the escape behavior of WT (black) and NaVDKO (red) fish. (C) Maximum neck angles of WT and NaVDKO fish. The maximum angle within 40 ms of the start of the response is plotted (n = 20 for WT and n = 18 for NaVDKO). (D) Maximum turn speed in WT and NaVDKO fish (n = 20 for WT and n = 18 for NaVDKO). The numerical data presented in this figure can be found in S1 Data.
(TIFF)

**S9 Fig. Swimming capability of adult fish assessed by the swimming treadmill.** (A) Schematic of swimming treadmill. (B) Swimming treadmill protocol. The flow rate was initially set at 15 cm/s for 1 min, followed by increments of 1 cm/s every min, starting at 20 cm/s. (C) Critical swimming speed (Ucrit) of individual fish plotted against the BMI. Open and

closed circles represent WT and NaVDKO fish, respectively (n = 20 for WT and n = 16 for NaVDKO). Dotted line represents regression line. Pearson's correlation coefficient was −0.35. (D) Comparison of Ucrit values between WT and NaVDKO fish (n = 20 for WT and n = 16 for NaVDKO). The numerical data presented in this figure can be found in S1 Data.
(TIFF)

**S10 Fig. Mathematical simulation of membrane potential.** (A and B) Membrane potentials with fiber lengths (L) of 500 (A) and 1,000 μm (B). Potentials were calculated based on the geometry shown in Fig 4A. The numerical data presented in this figure can be found in S1 Data.
(TIFF)

**S11 Fig. Simulation of membrane potential and the voltage dependency of the DHPR.** (A) Maximum potentials at the center of each region in the model shown in Fig 4F. The maximum amplitudes were well above 50 mV in all regions. (B) "Gating" current of DHPR in WT fish. The charge movement of the OFF current plotted against the membrane potential. Data shown as mean ± SEM (n = 5) and fitted with the Boltzmann equation. The inset shows representative traces of the "gating" current. Traces evoked by voltage steps ranging from −90 mV to 20 mV in 10 mV increments. The vertical and horizontal scale bars indicate 0.5 nA and 5 ms, respectively. The numerical data presented in this figure can be found in S1 Data.
(TIFF)

**S12 Fig. Membrane potentials calculated using squid, lobster, and goldfish parameters.** (A, B, and C) Membrane potential at the center (0.5 L) of region #6 (Fig 4F) calculated using the parameters obtained from the squid giant axon (A), lobster leg axon (B), and goldfish Mauthner axon (C). The resistivity values are shown in S7 Table. The mean values were used for the simulation. The panels above the plots indicate the timing of the synaptic input. The numerical data presented in this figure can be found in S1 Data.
(TIFF)

**S13 Fig. BTX staining and simulation in adult myofibers.** (A) Representative images of adult WT myofibers. Scale bar: 200 μm. Arrowheads indicate the linear ACh release sites. (B) Simulation of adult fibers. Total length of the fibers was approximately 1,100 μm, with ACh release site numbers ranging from 11 to 139. The current amplitude at the individual ACh release sites was set to 1.2 nA. (C) Membrane potential at the center of the fiber in each model. The square pulses above the plots indicate the timing of the synaptic input. The numerical data presented in this figure can be found in S1 Data.
(TIFF)

**S1 Table. Voltage-gated sodium channels in human and zebrafish.** Annotations and nomenclatures were derived from the Ensemble release113.
(DOCX)

**S2 Table. Summary of the RNA sequencing.**
(XLSX)

**S3 Table. DEG analysis of the trunk muscle of WT and NaVDKO fish.**
(CSV)

**S4 Table. DEG analysis of NaV alpha subunits.**
(XLSX)

**S5 Table. A1-subunits of voltage-gated calcium channels in human and zebrafish.**
(DOCX)

**S6 Table. DEG analysis for CaV alpha1 subunits.**
(XLSX)

**S7 Table. Diameter and resistivities of neurons and muscle fibers.** Average values are shown in brackets. Note that the specific leakage conductance is inverse of the resistivity of the membrane.
(DOCX)

**S1 Data. Numerical values for Figs 1A, 1B, 1C, 1D, 2B, 2C, 2D, 3B4, 3C4, 4B1, 4B2, 4D, 4E, 4G, S2C, S3B, S3C, S3D, S5A3, S5A4, S5B3, S5B4, S5C3, S5C4, S6C, S6D, S8B, S8C, S8D, S9B, S9C, S9D, S10A, S10B, S11A, S11B, S12A, S12B, S12C and S13C.**
(XLSX)

**S1 Movie. Escape response of 4-dpf WT embryos.**
(AVI)

**S2 Movie. Escape response of the 4-dpf NaVDKO embryos.**
(AVI)

**S3 Movie. Escape response of 4-dpf WT embryos in methylcellulose.**
(AVI)

**S4 Movie. Escape response of 4-dpf NaVDKO embryos in methylcellulose.**
(AVI)

**S5 Movie. In vivo calcium imaging of 5-dpf WT embryos.**
(AVI)

**S6 Movie. In vivo calcium imaging of the 4-dpf NaVDKO embryos.**
(AVI)

**S7 Movie. Calcium imaging of isolated fibers from 3-dpf WT embryos.**
(AVI)

**S8 Movie. Calcium imaging of isolated fibers from 5-dpf WT embryos in the presence of 1 µMTTX.**
(AVI)

**S9 Movie. Escape response of adult WT fish in methylcellulose.**
(AVI)

**S10 Movie. Escape response of adult NaVDKO fish in methylcellulose.**
(AVI)

## Acknowledgments

We thank Prof. Koichi Kawakami (National Institute of Genetics) for providing us with the pT2RUASGCaMP7a plasmid. We also thank Prof. Hiromi Hirata (Aoyama Gakuin University) for giving us advice on the swimming treadmill of zebrafish and critical reading of the manuscript. We are grateful to Ms. Natsuko Okuda and Ms. Risa Ishihara for their care of the zebrafish colonies.

## Author contributions

**Conceptualization:** Souhei Sakata, Fumihito Ono.

**Data curation:** Chifumi Akiyama, Souhei Sakata.

**Funding acquisition:** Souhei Sakata, Fumihito Ono.

**Investigation:** Chifumi Akiyama, Souhei Sakata.

**Project administration:** Souhei Sakata, Fumihito Ono.

**Supervision:** Fumihito Ono.

**Writing – original draft:** Souhei Sakata, Fumihito Ono.

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
