## [Editor Report · Decision Letter 0]

10 Feb 2025

Dear Dr Sakata,

Thank you for submitting your revised manuscript entitled "Normal locomotion in zebrafish lacking sodium channel" for consideration as a Research Article by PLOS Biology.

Your revisions have now been evaluated by the PLOS Biology editorial staff, and I'm writing to let you know that we would like to send your submission out for re-review.

IMPORTANT: Please could you include a "track changes" version of the manuscript when you upload your additional metadata (see next paragraph)?

Once your full submission is complete, your paper will undergo a series of checks in preparation for re-review. After your manuscript has passed the checks it will be sent out for review. To provide the metadata for your submission, please Login to Editorial Manager (https://www.editorialmanager.com/pbiology) within two working days, i.e. by Feb 12 2025 11:59PM.

Kind regards,

Roli Roberts

Roland Roberts, PhD

Senior Editor

PLOS Biology

rroberts@plos.org

---

## [Decision Letter · Decision Letter 1]

19 Mar 2025

Dear Dr Sakata,

Thank you for your patience while we considered your revised manuscript "Normal locomotion in zebrafish lacking sodium channel" for publication as a Short Report at PLOS Biology. This revised version of your manuscript has been evaluated by the PLOS Biology editors, the Academic Editor, the original reviewers and (at the request of the Academic Editor) one new reviewer, reviewer #4.

Based on the reviews, we are likely to accept this manuscript for publication, provided you satisfactorily address the remaining points raised by reviewer #4 and the following data and other policy-related requests.

IMPORTANT - please attend to the following:

a) Please change your Title to something more declarative and informative. We suggest: "Normal locomotion in zebrafish lacking the sodium channel NaV1.4 suggests that the need for muscle action potentials is not universal"

b) Please address the minor requests from reviewer #4.

c) Please address my Data Policy requests below; specifically, we need you to supply the numerical values underlying Figs Figs 1ABCD, 2BCD, 3B4C4, 4B1B2DEG, S2C, S3BCD, S4AB, S5A3A4B3B4C3C4, S6CD, S8BCD, S9BCD, S10AB, S11AB, S12ABC, S13C, either as a supplementary data file or as a permanent DOI’d deposition.

d) Please cite the location of the data clearly in all relevant main and supplementary Figure legends, e.g. “The data underlying this Figure can be found in S1 Data” or “The data underlying this Figure can be found in https://zenodo.org/records/XXXXXXXX

e) Thank you for providing the NEURON models. Please make any additional custom code available, either as a supplementary file or as part of your data deposition.

We expect to receive your revised manuscript within two weeks.

*Published Peer Review History*

*Press*

Sincerely,

Roli Roberts

Roland Roberts, PhD

Senior Editor

rroberts@plos.org

PLOS Biology

DATA POLICY:

Regardless of the method selected, please ensure that you provide the individual numerical values that underlie the summary data displayed in the following figure panels as they are essential for readers to assess your analysis and to reproduce it: Figs Figs 1ABCD, 2BCD, 3B4C4, 4B1B2DEG, S2C, S3BCD, S4AB, S5A3A4B3B4C3C4, S6CD, S8BCD, S9BCD, S10AB, S11AB, S12ABC, S13C. NOTE: the numerical data provided should include all replicates AND the way in which the plotted mean and errors were derived (it should not present only the mean/average values).

CODE POLICY

We require the original, uncropped and minimally adjusted images supporting all blot and gel results reported in an article's figures or Supporting Information files. We will require these files before a manuscript can be accepted so please prepare and upload them now. Please carefully read our guidelines for how to prepare and upload this data: https://journals.plos.org/plosbiology/s/figures#loc-blot-and-gel-reporting-requirements

DATA NOT SHOWN?

REVIEWERS' COMMENTS:

Reviewer #1:

The authors have addressed my concerns in revision and altered the manuscript to make it much more accessible.

Reviewer #2:

The authors have worked to address the main concerns, and present a compelling (and surprising) conclusion that sodium channels are not required for normal locomotion.

Reviewer #3:

In the present revised manuscript the authors have addressed all my major points and included extensive new data.

I believe the paper is now stronger.

Reviewer #4:

In this manuscript Akiyama et al provide a well-designed study to show that skeletal muscle voltage gated Na channels (NaV1.4) are not necessary for white fiber, fast-twitch muscle activation. The authors generate double knockouts of Nav11.4a and 1.4b (scn4aa and scn4ab genes), record sodium currents from wild type and knockouts, perform Ca imaging, locomotor studies, modeling and immunohistochemistry to show that embryonic and adult zebrafish myofiber activation does not require voltage gated sodium channel activation to result in fast swimming movements.

These findings challenge widely held beliefs about the role of voltage gated sodium channels associated with skeletal muscle fibers by providing a single example of an organism that does not appear to require NaV1.4 for fast twitch muscle activity. The experiments are well done. The authors provide strong data to show that compensatory mechanisms do not occur in the double knockouts (NaVDKO) and overall, the data is consistent with the conclusions. The authors have addressed reviewer's comments very well.

I have a couple of minor comments as mentioned below.

Comments are listed below.

Comments:

1. Fig 4. Clearer labelling of part C would be appreciated. For example, while it is included in the legend, labelling the image as BTX would be helpful.

2. The Discussion is fine, but I am curious why this situation occurs in zebrafish. Is there anything known or understood about medaka or goldfish that might shed light on the zebrafish state? Evolutionarily, is there an expectation that something similar will or should occur in other small vertebrates or muscle fibers that are innervated in the same pattern as zebrafish?

---

## [Editor Report · Decision Letter 2]

1 Apr 2025

Dear Dr Sakata,

Thank you for the submission of your revised Short Reports "Normal locomotion in zebrafish lacking the sodium channel NaV1.4 suggests that the need for muscle action potentials is not universal" for publication in PLOS Biology. On behalf of my colleagues and the Academic Editor, Simon Hughes, I'm pleased to say that we can in principle accept your manuscript for publication, provided you address any remaining formatting and reporting issues. These will be detailed in an email you should receive within 2-3 business days from our colleagues in the journal operations team; no action is required from you until then. Please note that we will not be able to formally accept your manuscript and schedule it for publication until you have completed any requested changes.

Sincerely, 

Roli Roberts

Senior Editor

PLOS Biology

rroberts@plos.org